# Characterization of the Volatile Compounds of Onion with Different Fresh-Cut Styles and Storage Temperatures

**DOI:** 10.3390/foods11233829

**Published:** 2022-11-27

**Authors:** Guangmin Liu, Yaqin Wang, Liping Hu, Hongju He

**Affiliations:** 1Institute of Agri-Food Processing and Nutrition, Beijing Academy of Agriculture and Forestry Sciences, Beijing 100097, China; 2Beijing Key Laboratory of Fruits and Vegetable Storage and Processing, Beijing 100097, China; 3Key Laboratory of Vegetable Postharvest Processing of Ministry of Agriculture and Rural Areas, Beijing 100097, China

**Keywords:** onions (*Allium cepa* L.), HS-GC-IMS, HS-SPME-GC-MS, storage, shape, temperature

## Abstract

The flavor of fresh onion and its processed products is an important index with which to evaluate its quality. In this study, the highly volatile compounds of onion with different fresh-cut styles (bulb, ring, and square) and different storage temperatures (4 °C, 20 °C, and 25 °C) were characterized at the molecular level, focusing in particular on the volatile sulfur compounds. Headspace-gas chromatography-ion mobility spectrometry (HS-GC-IMS) and headspace solid-phase microextraction-gas chromatography−mass spectrometry (HS-SPME-GC-MS) were employed. A total of 14 highly volatile compounds were identified in onion samples by HS-GC-IMS, and the square sample contained more volatile components. (*E*,*E*)-2,4-heptadianal, ethyl acetate, 2-methyl-1-pentanol, 2-pentylfuran, propyl acetate, and 2,6-dimethylpyrazine were produced in the ring and square samples when stored at higher temperatures, while pentanal, 2-heptenal, hexanal were decreased after cutting. Simultaneously, 16 sulfur compounds were identified in onions by HS-SPME-GC-MS. The sulfur compounds profile of the bulbs was significantly different from that of the rings and squares at any temperature. When stored at a low temperature (4 °C), cutting onions into a ring or square shape produced more sulfur. However, at higher temperatures (20 °C and 25 °C), fresh-cutting decreased the sulfur concentration. The total content of sulfur compounds was higher in the same cut style stored at higher temperatures (20 °C or 25 °C). 2-Mercapto-3,4-dimethyl-2,3-dihydrothiophene and 2,4-dimethylthiophene were formed during storage; however, (*E*)-1-(prop-1-en-1-yl)-3-propyltrisulfane, 1-(1-(methylthio)propyl)-2-propyldisulfane, (*Z*)-1-(1-propenyldithio)propyl disulfide, dipropyl trisulfide, and methyl 1-(1-propenylthio)propyl disulfide were lost from all samples after storage.

## 1. Introduction

Onion (*Allium cepa* L.), as a flexible vegetable of the *Alliaceae* family, is widely consumed because of its special flavor and beneficial compounds. It has been reported that onion has many functions, such as antifungal, antiviral, anti-inflammatory, antihypertensive, antidiabetic, antiallergic, and hypolipidemic functions [1,2]. It contains cinnamic acid, caffeic acid, lauric acid, mustard acid, quercetin, etc. [3]. In addition, onion has its own unique aroma, which is not only a kind of delicious food but also can be used to cover up the fishy smell and off-flavor in food [4]. Because of the remarkable properties of onion, this ancient vegetable has gained more attention from consumers. Due to their different consumption modes, fresh onions can be supplied directly to consumers or processed into different forms for the convenience of consumers, such as flakes, rings, or fragments. However, the chemical structure of the bioactive compounds in onions during storing and processing after harvesting may change, leading to profound differences in their bioavailability. In addition, their flavor can also change significantly [5].

The flavor of fresh onion and its processed products is an important index by which to evaluate its quality standard [6,7]. The flavor is not only influenced by the variety and origin of raw materials but also differs greatly based on the processing and storage methods [8]. With the development of research and analytical technologies, more and more volatile compounds have been found in onions, which include thioethers, aldehydes, ketones, alcohols, hydrocarbons, and heterocyclic compounds [8]. For example, Cecchi et al. [9] used headspace solid-phase microextraction (HS-SPME) to combine gas chromatography-mass spectrometry (GC-MS) and comprehensive two-dimensional gas chromatography-time of the flight-mass spectrometer (GC × GC-TOF-MS) to characterize the volatile compounds in onion slices with different drying cycles. Fifty-three volatile compounds were identified from onion slices, including sulfur compounds, aldehydes, ketones, carboxylic acids, alcohols, esters, and furans. Among these, dipropyl disulfide and other disulfide and trisulfide compounds with the typical pungent aroma of onion are the most abundant. In addition, many studies have checked the differences in the chemical components of fresh onion, onion powder, and its essential oils [10,11,12,13]. However, few researchers have focused on the influence of the fresh-cut process and storage temperature on the change in volatile compounds.

The main aroma compounds in onions are sulfur-containing compounds [14]. When the onion bulb is open, the alliinase enzyme can catalyze the hydrolysis of S-alk(en)ly-L-cysteine sulfoxide to generate various sulfur-containing volatile compounds [15,16]. The sulfocompounds in onions not only correlate with the medicinal properties of onion but also contribute to its pungent aroma. Hence, the content of sulfur compounds is an important reference with which to measure the quality of onions, and some studies have thus paid special attention to the changes in sulfur compounds in onions [17,18,19]. For example, Gao et al. [17] compared the volatile sulfocompounds in fresh onions and onion powders through SPME extraction and the volatile sulfocompounds acquired by simultaneous distillation extraction (SDE) in onion oils. According to them, dipropyl disulfide and dispropyl trisulfide were the greatest numbers of compounds in fresh onions (taking up 68.41% to 93.13% of the total volatile sulfocompounds), while dimethyl sulfides and thiophenes were identified as the main compounds in onion oils. Li et al. [19] found that dimethyl disulfide, dipropyl disulfide, and undecane had higher content in bulbs infected with *B. cepacia*. Thus far, 31 volatile sulfur compounds have been identified in onions, such as dimethyl disulfide, methylpropyl disulfide, allyl methyl disulfide, methylpropenyl disulfide, isopropyl disulfide, dipropyl disulfide, allyl disulfide, 3,5-diethyl-1,2,4-trithiopentane, 4,6-diethyl-1,2,3,5-tetrathione, 2,4-dimethylthiophene, and 5,7-diethyl-1,2,3,4,6-pentathione. However, the changes that occur in the sulfur compounds in fresh onions under different storage conditions are not clear.

Allium plants have complex matrix systems and low contents of flavor components, and the flavor components can be easily lost or denatured in the process of extraction and separation. Common extraction and separation methods include steam distillation, simultaneous distillation extraction (SDE), solid-phase microextraction (SPME), solvent-assisted flavor evaporation (SAFE), and supercritical fluid extraction (SFE). Gas chromatography-mass spectrometry (GC-MS), gas chromatography-olfactometry (GC-O), and comprehensive two-dimensional gas chromatography time of flight mass spectrometry (GC × GC-TOFMS) are the main methods used to identify volatile compounds [8]. Among these, SPME combined with GC-MS is the technique that is most frequently used to test volatile compounds qualitatively and quantitatively in foods because of its easy operation and results in no damage to the sample. More recently, gas chromatography-ion mobility spectrometry (GC-IMS) has been increasingly used for aroma characterization due to its ability to distinguish the differences effectively and intuitively in aroma among different food products. Combining the excellent separation capacity of GC with the high sensitivity and fast response of IMS, GC-IMS improves the accuracy of qualitative analysis. This technology separates the ions under atmospheric pressure, using the ion mobility of detected substances in an electric field [20]. IMS is used in many different fields, such as environmental analysis [21], medical diagnosis [22], pharmaceutical analysis [23], food safety [24], etc. The combination of GC-MS and GC-IMS, utilizing the advantages of the two methods, can provide more information on onion aroma under different storage conditions comprehensively, reliably, and scientifically.

As a result, the aims of this study were to: (1) determine the volatile profiles and comparatively analyze the differences in onions with different fresh-cut styles by HS-GC-IMS and (2) characterize the sulfur compounds composition in onion samples with different fresh-cut styles and different storage temperatures by SPME-GC-MS.

## 2. Materials and Methods

### 2.1. Materials

Fresh onion (*Allium cepa* L.) samples were bought at a local supermarket (Beijing, China) in 2021.

### 2.2. Chemical Standards and Reagents

2-Butanone, 2-pentanone, 2-hexanone, 2-heptanone, 2-octanone, 2-nonanone, and serial n-alkanes (C_6_~C_26_) were bought at Sigma-Aldrich Chemical Co., Ltd. (Shanghai, China), with 99% purity.

### 2.3. Sample Preparation

Untreated fresh onions were set as the control group. Fresh onions were cut into different shapes, as shown in Figure 1, including whole bulbs, rings, and squares (1 cm). Then, these fresh-cut onions were stored in fresh-keeping boxes at different temperatures (4, 20, and 25 °C) for 2 days. Labels “1”, “2”, and “3” were used to represent the onion bulbs, rings, and squares stored at 4 °C. Labels “4”, “5”, and “6” were used to represent the onion bulbs, rings, and squares stored at 20 °C. Labels “7”, “8”, and “9” were used to represent the onion bulbs, rings, and squares stored at 25 °C. All samples were in storage for 2 days.

### 2.4. HS-GC-IMS

The onion sample (1.0 g) was loaded in a 20 mL glass bottle for headspace extraction. The GC-IMS instrument (FlavourSpec^®^, G.A.S. Gesellschaft für analytische Sensorsysteme mbH, Dortmund, Germany) was provided with a syringe and an automated headspace sampling instrument unit. The extraction method was conducted as follows: when the injection needle temperature was 85 °C, the loaded sample was incubated (incubation speed was 500 rpm) at 40 °C for 15 min, and finally 500 μL of headspace air sample was injected into the injection port. Using an FS-SE-54-CB-1 (15 m × 0.53 mm, 1 μm, Dürem, Germany) column at a constant temperature of 60 °C, the volatile compounds were isolated. The flow rate of the initial carrier gas (N_2_, with 99.9% purity) was 2 mL/min, being held for 2 min. The carrier gas flow rate was then raised to 10 mL/min within 8 min, then raised to 100 mL/min within 10 min, and finally to 150 mL/min within 10 min. The temperature of the drift tube in IMS was maintained at 45 °C, and the carrier gas flow rate stayed at 150 mL/min. When the ionization source and the drift voltage were set as 6.5 keV, a 3H (300 MBq) was used.

### 2.5. Headspace Solid-Phase Microextraction-Gas Chromatography-Mass Spectrometry (HS-SPME-GC-MS)

Five grams of onion sample was placed into a 20 mL screw-capped vial. Then the sample was incubated at 40 °C for 5 min and extracted for 30 min. An automatic headspace sampling system (MultiPurpose Sample MPS 2 with an SPME adapter, from Gerstel Inc., Mülheim, Ruhr, Germany) with a 50/30 μm DVB/CAR/PDMS fiber (Supelco, Inc., Bellefonte, PA, USA) was applied to extract volatile components. As a result, the SPME fiber was inserted into the headspace and absorbed at 40 °C for 40 min. After extraction, the loaded SPME fiber was immediately eliminated from the sample vial and inserted into the injection port of GC-MS at 250 °C for 5 min in order to conduct further analysis. The analysis of each extract was repeated three times.

GC-MS analysis was conducted on an Agilent 7890B gas chromatographic column which was equipped with an Agilent 5975 mass selective detector. In order to isolate and characterize the aroma compounds, an INNOWAX capillary column (30 m × 250 μm × 0.25 μm; Agilent Technologies, Santa Clara, CA, USA) was applied. The carrier gas was helium (99.999%), with a constant flow rate of 1.0 mL/min and an inlet temperature of 250 °C. The sample was injected in splitless mode. The temperature program was as follows: the initial oven temperature, staying at 50 °C for 5 min, was ramped to 160 °C at 3 °C/min and then held at that temperature for 3 min, later, was raised to 230 °C at 10 °C/min and maintained there for 2 min. The temperature of the transfer line was 250 °C. With ionization energy being 70 eV, the electron ionization mode (EI) was used. The ion source temperature was 250 °C, and the mass range was from m/z 45 to 500.

### 2.6. Qualitative Analysis of the Volatile Compounds

By comparing the mass spectra and authentic standards of the volatile compounds on the INNOWAX column with those of pure standards, the volatile compounds were identified [25].

Based on the retention index (RI) and drift time (RIP relative) of the detected compounds, compared to that of the standard in the database, the aroma compound by GC-IMS was identified. In order to avoid interference effects, the library database was established using a single standard. Using the laboratory analysis view (LAV) software for GC-IMS relative to a C_3_–C_9_
*n*-ketones reference mixture, the RI was calculated.

As this study focused on the effect of storage methods on onion flavor, the relative peak area ratio was used for quantitative analysis. For this, the relative content of each compound was the ratio of the peak area of each compound in onions stored in different shapes to the peak area obtained in untreated onions (control group).

### 2.7. Statistical Analysis

With the analytical software LAV (G.A.S., Dortmund, Germany), the RIs and drift times of the volatile compounds were processed. In addition, the compounds were identified by the qualitative software GC × IMS Library Search (built-in NIST2014, IMS database). The gallery plot plug-in in LAV was used to compare the GC–IMS fingerprints.

## 3. Results and Discussion

### 3.1. Highly Volatile Compounds Identified by GC-IMS

The volatiles in the sample of untreated fresh onion, onion bulbs, onion rings, and onion squares stored at 25 °C were analyzed by GC-IMS. The ion migration time and the position of the reactive ion peak (RIP) were normalized. The ordinate represents the retention time of the gas chromatography, the abscissa represents the ion migration time, and the vertical line at the abscissa 1.0 is the RIP peak. Each point on the right of RIP represented a volatile compound extracted from the samples. Color represented the signal intensity of the substance. White indicated a lower intensity, and red indicated a higher intensity. A total of 15 volatile compounds, including 6 aldehydes, 2 alcohols, 2 esters, 1 sulfur, 1 ketone, 1 pyrazine, and 1 furan compound, were identified in onion samples by GC-IMS, as shown in Table 1.

Aldehydes and alcohols were the most abundant compounds. They have a green odor, similar to cut grass, leaves, or wood. Aldehydes in onion included pentanal, hexanal, heptanal, octanal, (*E*)-2-heptenal, and (*E*,*E*)-2,4-heptadienal. Alcohols included 1-propanol and 2-methyl-1-pentanol. According to previous research [26], aliphatic aldehydes and alcohols are produced upon cutting or chewing plant tissue from the lipoxygenase pathway. Most aldehydes identified in foods are produced by the oxidation of unsaturated fatty acids. For example, (*E*)-2-heptenal and pentanal are derived from w-6 fatty acids, and 2,4-heptadienal is derived from w-3 fatty acids. In plants, this process can be enzymatic and is called the lipoxygenase (LOX) pathway. The LOX pathway is predominately active in the green organs of plants in response to wounding, and it also gives rise to the formation of volatiles with green odors [26].

Ethyl acetate and propyl acetate have been identified in onion. Esters with fruity aromas have been reported in almost all fruits and are derived from the fatty acid and amino acid pathways [26], respectively.

2,6-Dimethylpyrazine, dimethyl ketone and 2-pentylfuran, and dimethyl disulfide, were also identified by GC-IMS. Among them, dimethyl disulfide has a low odor threshold and a typical cabbage aroma, while 2,6-dimethylpyrazine has a roasty aroma. Furans are the products of the autoxidation of polyunsaturated fatty acids. 2-pentylfuran in onion can be formed from linoleic acid [26].

### 3.2. Highly Volatile Compounds Profiles of Onions under Different Fresh-Cut Styles by GC-IMS

The difference comparison model was applied to compare the aroma variety of untreated onion (control group) and onions with different cutting styles (bulb, ring, and square) stored at 4 °C, 20 °C, and 25 °C (Figure 2). A topographic plot of fresh onions without any treatment was selected as a reference, and a topographic plot of other samples was deducted from the reference. If the volatile compounds were consistent, the background after the deduction was white, while red indicated that the concentration of the substance was higher than that in the reference, and blue indicated that the concentration of the substance was lower than that in the reference. As shown in Figure 2a, most of the signals in the topographic plot of onions appeared within a range of 100 and 1000 s of retention time, and significant differences in the aroma profile were obtained according to the variation in signals and colors. Onions after fresh-cut into rings and squares showed obvious improvements in the signals of aroma compounds. Moreover, the square samples showed more volatile components compared to those in untreated onions (Figure 2a).

Principal component analysis (PCA) is built using signal strength to emphasize differences in volatile components of onions treated with different fresh-cut styles (Figure 2b). The distribution diagrams of the first two principal components confirmed by PCA are shown; the cumulative variance contribution rates were 39% and 22%. The control samples (untreated onions), all bulb samples, and the rings and squares samples stored at 4 °C were located at the negative semi-axis of the first principal component, while the ring and square samples stored at 20 °C and 25 °C were located at the positive semi-axis. The results revealed that the volatile compounds of all onion bulbs and ring- and square-cut samples stored at 4 °C are relatively similar to those of the control, while the volatile compounds of onion rings and squares stored at 20 °C and 25 °C are similar but quite different from those of the control.

Next, gallery plot analysis, as a fingerprinting technique, was used to show the differences in the volatile compounds of onions under different treatments more specifically and intuitively [27]. As shown in Figure 2c, each row shows all the signal intensities of one sample, and each column reveals the selected compound in different samples. Furthermore, colors represent the signal intensity (compound concentration) of the volatile molecules. The low intensities are expressed in white color, the high intensities in red, and the higher signal intensities in a deeper color. Accordingly, the differences in the volatile compounds in different onion samples were observed. Each sample had its own characteristic volatile components.

As shown in Figure 2c, most of the volatile compounds had higher concentrations when stored at a higher temperature. For instance, the contents of (*E*,*E*)-2,4-heptadianal were significantly higher in the square onion samples stored at 25 °C. Ethyl acetate, 2-methyl-1-pentanol, 2-pentylfuran, and propyl acetate had higher concentrations in ring and square samples stored at 20 °C and 25 °C. Octanal and 2,6-dimethylpyrazine had the highest concentration in ring samples stored at 20 °C. This result showed that these compounds were produced in the ring and square samples when stored at higher temperatures (20 °C and 25 °C). Dimethyl trisulfide had the highest concentration in bulb samples stored at 25 °C.

However, the contents of pentanal and 2-heptenal in the bulbs stored at 4 °C were higher than that stored at 20 °C and 25 °C. The contents of acetone and 1-propanol were lower in bulbs stored at 20 °C and 25 °C. It was revealed the contents of pentanal, 2-heptenal, acetone, and 1-propanol decreased after cutting and storage. Meanwhile, the content of heptanal in the rings and squares was higher than those in bulb samples. By contrast, hexanal had the lowest concentrations in square samples. These results showed that heptanal was produced after cutting, while hexanal was decreased after cutting.

### 3.3. Volatile Sulfur Compounds Characterized by GC-MS

Sulfur compounds with low odor thresholds are the most important factors that lead to the unique aroma of onions. However, most of them are heat-sensitive and easily volatile, meaning that it is very easy for them to be denatured in the process of pretreatment and subsequent processing, resulting in flavor changes. According to previous research, in onions, S-1-propenyl cysteine sulfoxide (CSOs) is the main precursor of sulfur-containing volatiles [28]. If onions are chopped, a typical aroma appears within seconds. Due to the disruption of the cells, the enzyme alliinase comes into contact with CSOs and cleaves the C(ˇ)-S bond, thus releasing ammonia, pyruvate, and a series of unstable sulfenic acids. These sulfenic acids yield, by purely chemical reactions, more stable volatiles, such as the well-known thiosulfinate allicin in garlic, and further volatiles, such as disulfides [26].

A total of 16 sulfur compounds were identified, such as propan-1-thiol, 2,4-dimethylthiophene, and dipropyl disulfide. The relative contents of all compounds are exhibited in Appendix A. According to the research by Sun [29], compounds with a propyl thiol group in their molecular structure generally have an onion-like flavor. Among these 16 compounds, the structure of 10 compounds contained propyl thiol, among which propyl mercaptan, dipropyl disulfide, and methyl propyl trisulfide were the most important onion characteristic species. Propan-1-thiol has a strong irritation effect and is the main source of the antibacterial effect of onion. Dipropyl disulfide and dipropyl trisulfide are the main volatile compounds present in fresh onions and are the main reason for the strong aroma of fresh onions [30]. In addition, dipropyl disulfide, (*E*)-1-(prop-1-en-1-yl)-3-propyltrisulfane, and 1-(1-(methylthio)propyl)-2-propyldisulfane have also been identified in onion oil [31]. Of note, 2,4-dimethylthiophene, 1-allyl-2-isopropyldisulfane, 2-mercapto-3,4-dimethyl-2,3-dihydrothiophene, 1-(1-(methylthio)propyl)-2-propyldisulfane, 1-methyl-2-(1-(propylthio)propyl)disulfane, 2-methoxy-5-methyl-thiophene, methyl 1-(1-propenylthio)propyl disulfide, trans-3,6-diethyl-1,2,4,5-tetrathiane, 6-ethyl-4,5,7,8-tetrathiaundecane, and (*Z*)-1-(1-propenyldithio)propyl propyl disulfide were identified in onion for the first time.

### 3.4. Effect of Storage Temperature and Fresh-Cut Style on Sulfur Compounds in Onion

In order to analyze the correlation between the content of each sulfur compound and the sample, a hierarchical clustering analysis was conducted. As depicted in Figure 3, the sulfur compounds profile of the bulbs was significantly different from that of the rings and squares stored at any temperature, while the ring and square samples stored at the same temperature were clustered into one cluster. It was indicated that fresh-cut styles and storage temperature of onion had a great influence on their release of sulfur compounds, and different onion samples can be distinguished according to the content of sulfur compounds.

For the same storage temperature, as shown in Appendix A, at 4 °C, the total relative content of sulfur compounds was the lowest in the bulb samples and the highest in the square samples. However, at 20 °C and 25 °C, the content of sulfur compounds was the highest in the bulb samples but lowest in the ring samples. These results revealed that at a low temperature (4 °C), cutting onions into a ring or square shape will produce more sulfur. However, at higher temperatures (20 °C and 25 °C), fresh-cutting decreased the sulfur concentration.

For the same cut style, as shown in Appendix A, the total content of sulfur compounds in bulb samples at 20 °C and 25 °C was significantly higher than that after storage at 4 °C. The total content of sulfur compounds in onion rings was the highest at 20 °C, followed by 4 °C and finally 25 °C. The content of sulfur compounds in squared onion was the highest at 25 °C but lowest at 4 °C. These results indicated that when storing at the same cut style, the total content of sulfur compounds was higher in the sample stored at higher temperatures (20 °C or 25 °C)**.**

To more accurately find the difference between different samples, the content change of each compound was further analyzed. As shown in Figure 4, the patterns of their changes in concentration could be divided into six categories: (1) The concentrations of propan-1-thiol (Figure 4a) and dipropyl disulfide (Figure 4b) were higher in whole bulbs at 4 °C and 20 °C, but significantly lower in the bulb samples stored at 25 °C. However, the concentrations in rings and squares were lower in the samples stored at 4 °C and 20 °C but higher in the samples stored at 25 °C. (2) (*E*)-1-(prop-1-en-1-yl)-2-propyldisulfane, (*E*)-1-(prop-1-en-1-yl)-3-propyltrisulfane, 1-(1-(methylthio)propyl)-2-propyldisulfane, (*Z*)-1-(1-propenyldithio)propyl propyl disulfide, and dipropyl trisulfide had the highest concentrations in the bulb samples stored at 20 °C and lowest concentrations in rings and squares samples stored at 4 °C, and the concentrations in rings and squares increased along with the increase in storage temperatures (Figure 4c–g). (3) Methyl 1-(1-propenylthio)propyl disulfide, 2-methoxy-5-methyl-thiophene, 6-ethyl-4,5,7,8-tetrathiaundecane, and 1-methyl-2-(1-(propylthio)propyl)disulfane had the highest concentrations in the whole bulb samples stored at 20 °C. However, for rings and squares, their concentrations decreased along with the increase in storage temperature (Figure 4h–k). (4) The concentrations of methyl propyl trisulfide increased along with the increase in storage temperatures (Figure 4l) in all samples. (5) The concentrations of trans-3,6-diethyl-1,2,4,5-tetrathiane, 2-mercapto-3,4-dimethyl-2,3-dihydrothiophene, and 2,4-dimethylthiophene in bulbs were increased along with the increase in storage temperature, while the concentrations of trans-3,6-diethyl-1,2,4,5-tetrathiane and 2-mercapto-3,4-dimethyl-2,3-dihydrothiophene in rings and squares stored at 20 °C were higher than those stored at 4 °C and 25 °C, while the concentrations of 2,4-dimethylthiophene in the square samples decreased along with an increase in temperature (Figure 4m–o). (6) 1-allyl-2-isopropyldisulfane had the lowest concentration in all samples stored at 20 °C (Figure 4p).

Of note, compared with the control groups, most of the sulfur compounds were decreased after cutting, including propan-1-thiol (Figure 4a), (*E*)-1-(prop-1-en-1-yl)-3-propyltrisulfane (Figure 4d), 1-(1-(methylthio)propyl)-2-propyldisulfane (Figure 4e), (*Z*)-1-(1-propenyldithio)propyl propyl disulfide (Figure 3f), dipropyl trisulfide (Figure 4g), and methyl 1-(1-propenylthio)propyl disulfide (Figure 4h). It was revealed that these compounds were lost after cutting. By contrast, the concentrations of 2-mercapto-3,4-dimethyl-2,3-dihydrothiophene and 2,4-dimethylthiophene were increased after cutting at any temperature. It was revealed that these compounds were produced after fresh cutting.

## 4. Conclusions

In this study, fresh onions were cut into three shapes—i.e., onion bulbs, onion rings, and onion squares—and then stored at different temperatures (4 °C, 20 °C, and 25 °C).

The highly volatile compounds in onions and their corresponding differences were identified and analyzed by HS-GC-IMS. At the same time, the sulfur compounds stored in different fresh-cut style onions at different temperatures were identified and compared using HS-SPME-GC-MS. A total of 14 easily volatile compounds and 16 sulfur compounds were identified as quality markers.

It was the first study to explore the changes that occur in the volatile compounds, especially sulfur compounds, in fresh onions under different cutting styles and different storage conditions. These results provide useful information for the quality control of onion flavor, which is conducive to improving the storage of fresh onions. For example, the concentration of these compounds can reflect the quality of the sample to a certain extent, especially the flavor quality. In this study, the content differences of different compounds under each condition in samples have been clarified. Therefore, based on these results, the onion storage conditions could be selected and adjusted scientifically. Of course, the specific contribution of each compound to the specific aroma of onion needs to be further investigated in the future.

## Figures and Tables

**Figure 1 foods-11-03829-f001:**
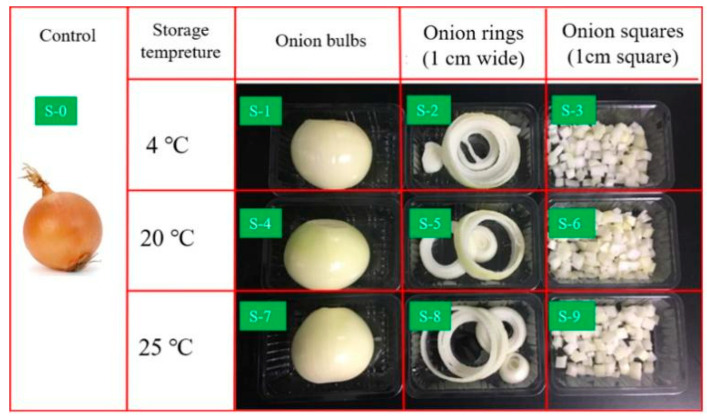
Appearance of samples. Labels: “S-0” represents the untreated onion (control group); “S-1”, “S-2”, and “S-3” represent the onion bulbs, onion rings, and onion squares stored at 4 °C; “S-4”, “S-5”, and “S-6” represent the onion bulbs, onion rings, and onion squares stored at 20 °C; “S-7”, “S-8”, and “S-9” represent the onion bulbs onion rings, and onion squares stored at 25 °C.

**Figure 2 foods-11-03829-f002:**
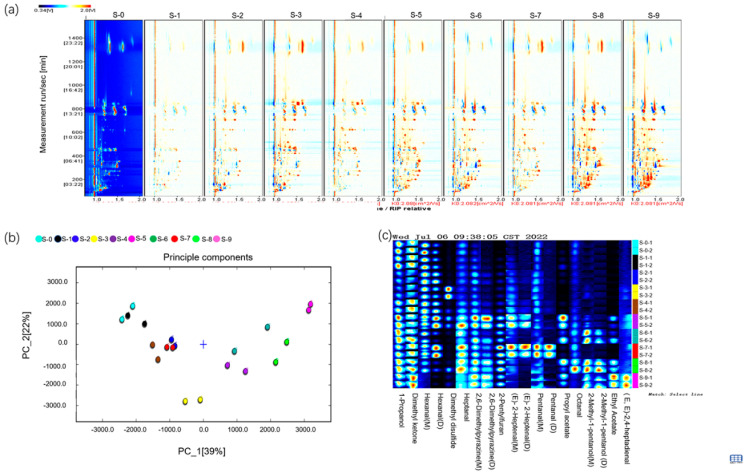
(**a**) GC-IMS spectra of volatile organic compounds in onion samples. (**b**) Principal component analysis (PCA) diagram of onion samples. (**c**) Gallery plot (fingerprint) of volatile organic compounds in onion samples. Labels: “S-0” represents the untreated onion (control group); “S-1”, “S-2”, and “S-3” represent the onion bulbs, onion rings, and onion squares stored at 4 °C; “S-4”, “S-5”, and “S-6” represent the onion bulbs, onion rings, and onion squares stored at 20 °C; “S-7”, “S-8”, and “S-9” represent the onion bulbs onion rings, and onion squares stored at 25 °C.

**Figure 3 foods-11-03829-f003:**
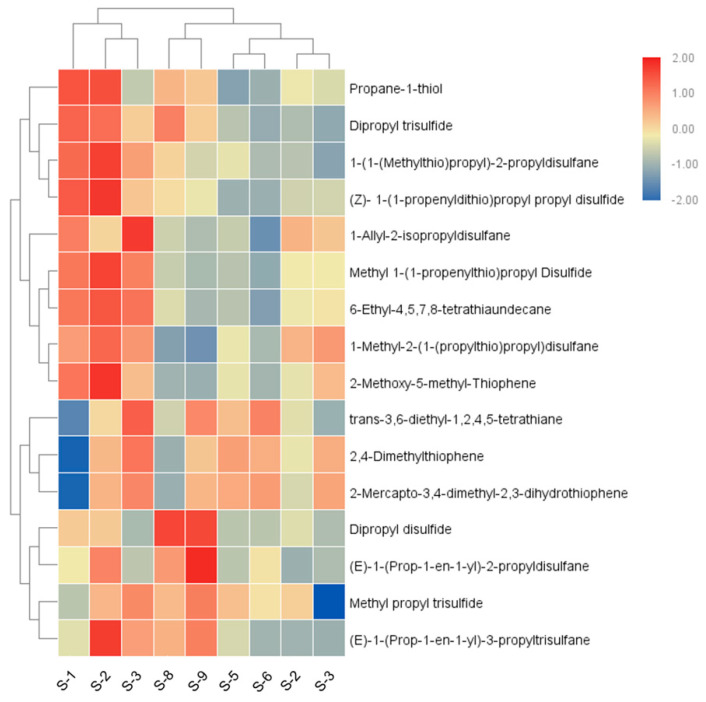
Clustering heat map of volatile sulfur compounds in each sample. “S-1”, “S-2” and “S-3” represent the onion bulbs, onion rings, and onion squares stored at 4 °C; “S-4”, “S-5” and “S-6” represent the onion bulbs, onion rings, and onion squares stored at 20 °C; “S-7”, “S-8” and “S-9” represent the onion bulbs onion rings, and onion squares stored at 25 °C.

**Figure 4 foods-11-03829-f004:**
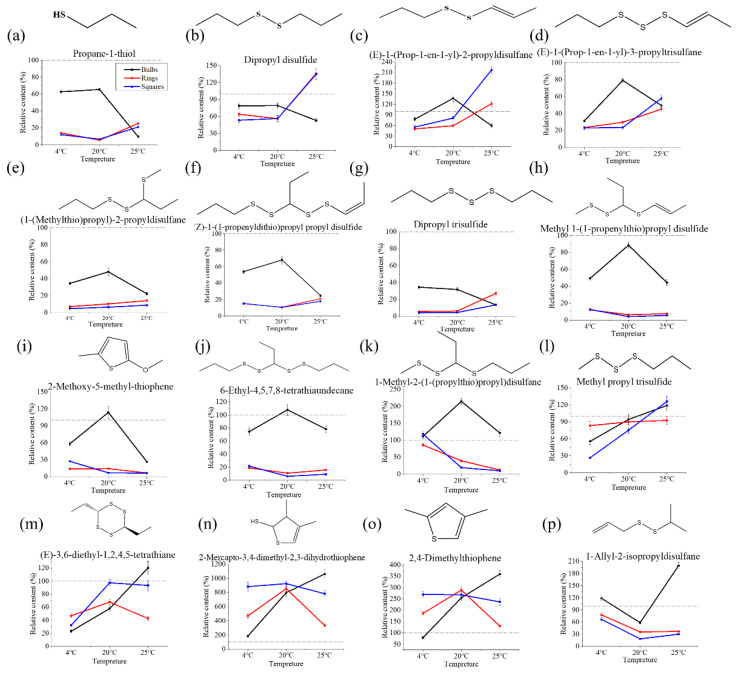
Concentration changes of 16 volatile sulfur compounds in onions with different fresh-cut styles and different storage temperatures. (**a**–**p**), the structure and relative contents of propane-1-thiol, dipropyl disulfide, (*E*)-1-(prop-1-en-1-yl)-2-propyldisulfane, (*E*)-1-(prop-1-en-1-yl)-3-propyltrisulfane, 1-(methylthio)propyl)-2-propyldisulfane, (*Z*)-1-(1-propenyldithio)propyl propyl disulfide, dipropyl trisulfide, methyl 1-(1-propenylthio)propyl disulfide, 2-methoxy-5-methyl-thiophene, 6-ethyl-4,5,7,8-tetrathiaundecane, 1-methyl-2-(1-(propylthio)propyl)disulfane, methyl propyl trisulfide, (*E*)-3,6-diethyl-1,2,4,5-tetrathiane, 2-mercapto-3,4-dimethyl-2,3-dihydrothiophene, 2,4-dimethylthiophene, 1-allyl-2-isopropyldisulfane the relative contents of, respectively. The grey dotted line was the relative value of the control groups (100).

**Table 1 foods-11-03829-t001:** The volatile compounds identified by headspace-gas chromatography-ion mobility spectrometry.

Count	Compound	Formula	MW ^a^	LRI ^b^	Rt [sec] ^c^	Dt [a.u.] ^d^	Comment
1	Pentanal	C5H10O	86.1	699.2	191.218	1.18096	Monomer
2	Pentanal	C5H10O	86.1	699.2	191.218	1.43014	Dimer
3	Hexanal	C6H12O	100.2	796.3	275.615	1.25142	Monomer
4	Hexanal	C6H12O	100.2	794.1	273.480	1.56955	Dimer
5	Heptanal	C7H14O	114.2	900.6	401.145	1.33159	
6	Octanal	C8H16O	128.2	1015.1	608.637	1.40019	
7	(*E*)-2-Heptenal	C7H12O	112.2	961.3	505.127	1.26142	Monomer
8	(*E*)-2-Heptenal	C7H12O	112.2	961.1	504.710	1.67421	Dimer
9	(*E*,*E*)-2,4-heptadienal	C7H10O	110.2	1005.5	592.246	1.19157	
10	1-Propanol	C3H8O	60.1	568.8	131.415	1.25890	
11	2-Methyl-1-pentanol	C6H14O	102.2	848.9	332.835	1.29958	Monomer
12	2-Methyl-1-pentanol	C6H14O	102.2	844.2	327.260	1.59906	Dimer
13	Ethyl Acetate	C4H8O2	88.1	617.2	150.447	1.34254	
14	Propyl acetate	C5H10O2	102.1	714.8	202.823	1.48425	
15	2,6-Dimethylpyrazine	C6H8N2	108.1	907.9	412.523	1.14445	Monomer
16	2,6-Dimethylpyrazine	C6H8N2	108.1	908.0	412.710	1.53201	Dimer
17	Dimethyl ketone	C3H6O	58.1	501.5	108.890	1.13033	
18	2-Pentylfuran	C9H14O	138.2	997.9	579.587	1.25453	
19	Dimethyl disulfide	C2H6S2	94.2	744.2	226.761	1.14486	

^a^ MW means Molecular weights; ^b^ LRI means linear retention indices; ^c^ Rt [sec] means retention time, and the unit of the retention time was second; ^d^ Dt [a.u.] means drift time.

## Data Availability

Data sharing is not applicable.

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
