# Peer review of "Characterization of the Volatile Compounds of Onion with Different Fresh-Cut Styles and Storage Temperatures"

_foods, 2022, doi:10.3390/foods11233829_

Round 1
Reviewer 1 Report
- Please correct: “they contain cinnamic acid, caffeic acid, lauric acid, mustard acid, quercetin, etc.” to “it contains cinnamic acid, caf- feic acid, lauric acid, mustard acid, quercetin, etc.”
- The seventh line in the first paragraph is missing. Please check and complete.
- Sample collection: Please mention, in plant material part, the year of plant collection.
- The number of atoms in chemical formula “C6-C26 and N2” should be indicated by a subscript.
- On which bases the author choosed the storage temperature and duration.
- How authors choose the conditions of sample preparation, chromatography separation and detection? Did authors optimize these conditions?
- Please add the meaning of the shortened word used in table 1.
- Please use the same sample name reference either
- Authors used three different sample references: "1, 2, 3, ...etc.", "S1, S2 , S3, ... etc.", and "bulb 4, bulb 20, bulb 25, ...etc". Please use the same sample name through the manuscript.
- Please correct “Propane-1-thiol” to “Propan-1-thiol”.
- Did authors have any explanation about the variations of volatile compounds according to the temperature of storge or the cutting process?
Reviewer 2 Report
The manuscript entitled: “Characterization of the volatile compounds of onion with different fresh-cut styles and storage temperatures” reports experimental results of a study on volatile compounds of onion with different fresh-cut styles (namely: bulb, ring, and square) considering also different storage temperatures. The manuscript fits the scope and aims of the Journal. There are however a few points to clear, e.g the phrasing in the Introduction first lines, where it seems that some words are missing in the text. Please explain better what it is meant by “quality standard” in the Introdution and substantiate this sentence better for clarity with reference to the context. The novelty and contribution to the area of interest of the manuscript and the limits shoud be detailed considering that there is wide literature existing on the topic. The experimental section is properly assessed, nonetheess please detail what do is meant by “control group”, since all the samples are fresh untreated. Please comment on this point. How many sampls have been studied? Please specify. The correlation between “quality control of onion flavor, which is conducive to improving the storage of fresh onion” as mentioned in the Conclusion section should be better substantiated and assessed for clarity. It is suggested to avoid repetitions in the Conclusion section and set the end points and persective possible application in the area of interest.
Reviewer 3 Report
The work is interesting, but requires corrections and changes to the text listed below:
- Introduction - editorial correction
- Page 2 - 2 paragraph - B. cepacia should be in italics
- Page 4 - the authors provide divergent data on sorption conditions. “Five grams of onion sample was placed into a 20 mL screw-capped vial, then the sample was incubated at 50 °C for 5 min and extracted for 30 min. …… Subsequently, the SPME fiber was inserted into the headspace and adsorbed for 40 min at 45 °C.”
- Page 4 contains the abbreviation MBFB, the meaning of which has not been explained in the text of the publication.
- Table 2 - the authors did not indicate which of the given ions is a molecular ion, nor did they specify intensity of ions. The table also lacks the determined values of retention indexes that would allow the verification of the determined compounds. Below the table there is an explanation of the symbol S which was not used in the table in the column marked identifi.
- Page 10 - figure 4 is too small and therefore illegible.
Reviewer 4 Report
Please carefully check the spelling and grammar within the entire manuscript. English language style should be improved.
There are some different Fonts in the table compared to the rest of the document.
Table 2 is difficult to follow as is not well-organized. Please check the table notes as there are unuseful details mentioned there. What do you mean to relative content? Explain near the table title or in the table notes.
The resolution of Fig.2 and Fig. 4 must be adjusted.
The authors should address more the significance of their findings. How could these results impact the food industry?
Reviewer 5 Report
Subject of this manuscript is very interesting. The experimental design needs to be improved. Manuscript needs minor improvement:
How many repetitions of the HS-SPME – GC-MS analysis were made?
The authors do not mention significant or insignificant differences when presenting the results (it should be presented with addition of a standard deviation).
The authors should calculate and add in Table 1. linear retention indices (LRI) of aroma compounds instead retention time.
Round 2
Reviewer 4 Report
I recommend the paper acceptance after the English language improvement. There are still some grammar errors. Please check carefully.